# Coupling Hydrological and Hydrodynamic Models for Assessing the Impact of Water Pollution on Lake Evaporation

**Janine Brandão de Farias Mesquita [1],\* and Iran Eduardo Lima Neto [2]**

1   Campus Crateús, Federal University of Ceará—UFC, Crateús 63708-825, Brazil
2   Department of Hydraulic and Environmental Engineering, Federal University of Ceará—UFC, Fortaleza 60020-181, Brazil
\*   Correspondence: janine@crateus.ufc.br; Tel.: +55-88-3691-9700

**Abstract:** The present study evaluated the impact of hydrological variability on the hydrodynamics of an urban lake in Brazil, considering water quality dynamics and its effects on evaporation. The Storm Water Management Model (SWMM) was applied to the lake basin, and the two-dimensional model CE-QUAL-W2 was used to simulate the hydrodynamics and lake evaporation. The two models were coupled to carry out the integrated basin-lake modeling. Then, two water quality models were applied: a transient complete mixing model and an empirical model based on wind speed. Time series of total phosphorus (TP) were generated, and empirical correlations between TP and hydrological variables were proposed. Modeled TP and measured biochemical oxygen demand (BOD) were correlated with monthly Class A pan coefficients (K) adjusted for the lake. The K-values were negatively correlated with TP modeled by the complete mixing model ($R^2 = 0.76$) and the empirical model ($R^2 = 0.52$), as well as by BOD measurements ($R^2 = 0.85$). This indicates that water pollution attenuates evaporation rates. Scenarios of lake pollution and level reduction due to evaporation were also analyzed. The results from this study are important to improve the management of lakes and reservoirs by including the impact of pollution on the water balance.

**Keywords:** hydrodynamic modeling; SWMM; CE-QUAL-W2; Class A pan coefficient; water quality; evaporation rates

## 1. Introduction

Changes in the physical state of water, which make up the hydrological cycle, are an important indicator of the environmental characteristics of watersheds. Changes in land use are key factors in the urbanization process [1,2]. In this case, the removal of vegetation cover and soil sealing have been identified as major elements responsible for altering the hydrological cycle [1].

Several impacts on hydrology can be directly related to unplanned urbanization [2–5], such as increased peak flow and the decrease in the time of concentration of the basins. In addition, impacts resulting from land use and occupation can directly affect receiving water bodies [6,7]. These may be subject to changes in their hydrodynamic characteristics, which, in turn, directly influence water quality [5]. Therefore, water quality can affect the energy balance of water bodies [8,9] and, consequently, evaporation [10]. In this sense, it is necessary to carry out studies that consider the complex dynamics involved in the basin-receiving water body interaction.

Studies related to hydrological-hydraulic modeling and simulation of urban drainage systems are essential for the representation of physical processes in urban watersheds [2,11–13]. Furthermore, computer modeling enables the analysis of various scenarios, optimizing costs and allowing the proposition of mitigating measures [2]. In this way, modeling tools can support decision makers.

Additionally, the hydrodynamic modeling of lakes and reservoirs allows the analysis of circulation patterns and the various forcings to which water bodies are subjected [14,15].

This makes it possible to assess impacts and understand physical, chemical, and biological variables [16,17]. In this sense, integrated basin-lake modeling can be an essential tool to analyze different processes and responses in both systems (basin and lake). Therefore, the integrated analysis of the hydrological characteristics of the basin and hydrodynamics of the lake, considering the water quality, would allow a holistic assessment of the physical parameters of the water body, notably evaporation.

In the literature, a variety of studies have addressed the impact of land use and occupation (e.g., [1]), as well as climate change [18], on the hydrological characteristics of basins and receiving water bodies. Some have studied the impact of climate on hydrodynamics, including the thermal stratification patterns of lakes [19–23]. Others estimated pollutant loads and modeled water quality in different environments [2,13,24]. Additionally, several studies have coupled hydrological models to hydrodynamic models [21,25,26]. However, to the authors' knowledge, the coupling of the Storm Water Management Model—SWMM [27] to the CE-QUAL-W2 model [28] has not been performed in previous studies. This would provide a robust tool with relatively low computational effort for the integrated analysis of the interaction between the watershed and the receiving water body, facilitating the understanding of the processes and assessment of the basin's impacts on the water body. It is known that urban watersheds are more subject to intense hydrological variability [2]. This can have a more severe impact on the receiving water bodies, both from the point of view of hydrodynamics and water quality. Thus, a hypothesis can be raised: shallow water bodies in urban areas would be more susceptible to changes in their hydrodynamic characteristics, depending on the interaction with their contribution watershed. It is suggested that coupling hydrological, hydrodynamic, and water quality models allows evaluation of the impact of hydrology on lake hydrodynamics, as well as the dynamics of water quality in the basin-lake system and its impact on evaporation rates. In this line, no studies were identified that combined hydrological and hydrodynamic models to understand the basin-lake interaction in urban areas, as well as to assess the impact of hydrological variability on the hydrodynamic characteristics of the lake.

Regarding evaporation, a recent study suggested that water quality would possibly affect this process in lakes [10]. The present paper advanced in this direction, as it expanded that assessment by applying water quality models for total phosphorus, considering the interannual and seasonal hydrological variability, as well as the coupling of a hydrological-hydraulic model to a two-dimensional hydrodynamic model. Furthermore, in addition to total phosphorus, in this paper, the impact of water quality on evaporation rates was also evaluated, considering the biochemical oxygen demand (BOD) as a reference parameter.

In this sense, this work intends to advance in relation to the current state of the art by proposing a hydrodynamic modeling tool integrating basin to lake. The objective is to evaluate the impact of hydrological characteristics on hydrodynamics, considering water quality and its impact on evaporation rates. Thus, the following innovative aspects can be highlighted: (I) the coupling of a one-dimensional hydrological-hydraulic model to a two-dimensional hydrodynamic model; (II) the assessment of the impact of hydrological variability on shallow lake hydrodynamics; and (III) the assessment of the impact of water quality, including the effects of both phosphorus and BOD, on evaporation rates in a lake.

With this approach, the aim is to understand in an integrated way the hydrological, hydrodynamic, and water quality processes. Hence, it is intended to apply tools that can support decision makers in proposing effective mitigation measures and environmental recovery, aiming to minimize the impacts of land use and occupation on water bodies.

## 2. Materials and Methods

### 2.1. Study Area

The study area is the Santo Anastácio Lake basin (outlet at coordinates: latitude −3.74 S, longitude −38.57 W), located in Fortaleza, Ceará, Northeast region of Brazil (Figure 1). It is an urban watershed that includes nine neighborhoods, namely: Amadeu Furtado, Bela Vista, Bonsucesso, Couto Fernandes, Demócrito Rocha, Jóquei Clube, Pici,

Parangaba, and Rodolfo Teófilo. It has an area of 611.27 ha and is predominantly uncharacterized by its natural composition. As the main watercourse, it has an urban drainage channel of approximately 2.5 km, with a rectangular section, an outlet from the Parangaba lake, located upstream, in addition to several urban drainage galleries that flow into that channel.

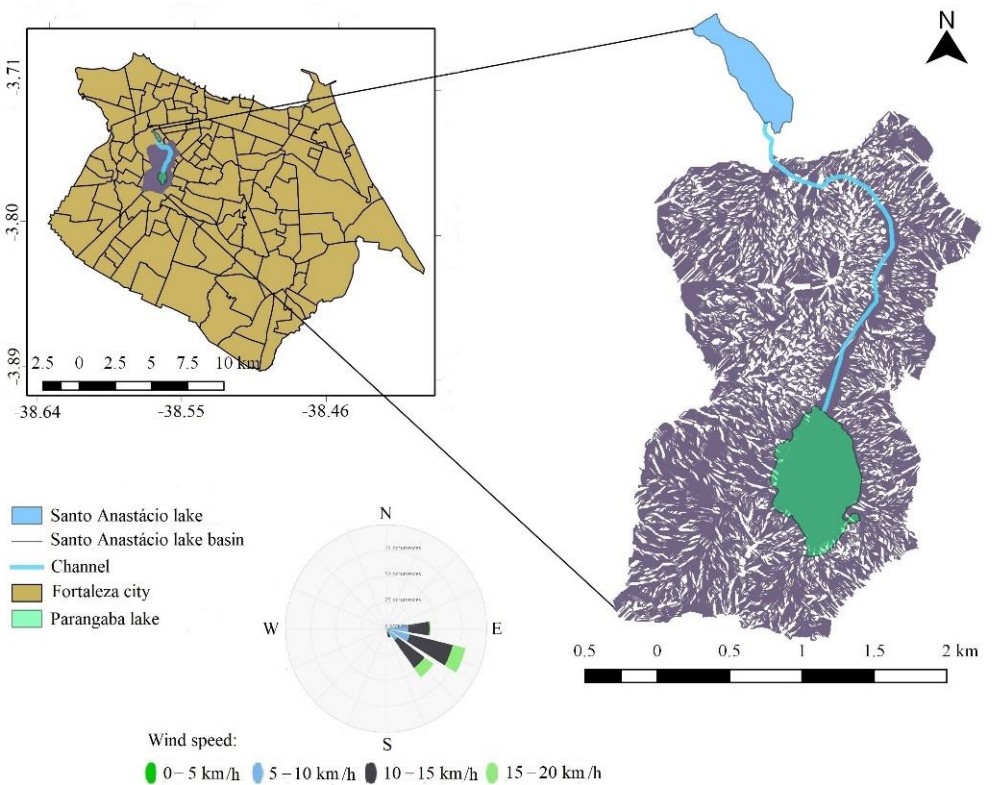

**Figure 1.** Contribution basin of the Santo Anastácio Lake inlet section in Fortaleza, Ceará, Brazil. Shapefile source (Fortaleza): Fortaleza (2020) and wind rose for a typical day: www.meteoblue.com (accessed on 3 September 2022). Datum: SIRGAS 2000, Zone: 24 S.

Santo Anastácio Lake is located in a coastal region, with an average annual rainfall of 1338 mm, concentrated predominantly from January to May. It has two defined seasons, characterized by seasonality (rainy and dry periods) and a hot tropical sub-humid climate, with an average temperature varying from 26 to 28 °C [29], a maximum of 30.1 °C and a minimum of 23.6 °C, average annual total evaporation of 1435.2 mm and average wind speed of 3.2 m·s$^{-1}$ [30].

### 2.2. Field Data and Study Framework

The meteorological data used in the present work were obtained from the meteorological station of the Pici campus of the Federal University of Ceará (UFC), located approximately 1 km from Santo Anastácio Lake, in the municipality of Fortaleza, state of Ceará, Brazil. The data obtained were precipitation, air temperature, wind speed and direction (at 10 m from the ground), evaporation from the Class A pan and Piché evaporimeter, relative humidity, and cloud cover, measured three times a day (9.00, 15.00, and 21.00). The measurements of flow rate were carried out from May to March 2013, 2014, 2018, and 2019, totaling 24 measurements. An electromagnetic propeller anemometer MiniWater20, from Omni Instruments, Templeton, CA, USA, (speed range 0.02–5.00 m·s$^{-1}$), was used for velocity measurements, and a ruler for water depth measurement. Velocity measurements were performed at four points of the channel cross-section, at heights of 0.2 and 0.8 m in relation to the channel bottom. From this information, it was possible to calculate the flow rates at the inlet and outlet of Santo Anastácio Lake. Additionally, the flow measurements

were conducted along the channel, close to the galleries, totaling three measurement points. The flow measurements along the channel were carried out in October 2013 (dry season), aiming to observe the hourly and longitudinal variation of sanitary sewage discharges into Santo Anastácio Lake. Concomitantly to the flow measurements, the temperature of the inflows and outflows of the lake was measured using a multiparametric probe (HI9820 Hanna Instruments, Woonsocket, RI, USA).

The collection of water samples was performed concomitantly with the measurements of the inflow and outflow of the lake. A van Dorn bottle was used for this purpose. The parameters analyzed were total phosphorus (TP) in 2013, 2014, 2018, and 2019 and BOD in 2013. TP analyzes were conducted in the Chemical Analysis Laboratory (LAQUIM), while BOD was analyzed at the Environmental Sanitation Laboratory (LABOSAN), both at the Federal University of Ceará (UFC). Water quality analyzes were performed according to Standards Methods [31]. The study framework is summarized in Figure 2.

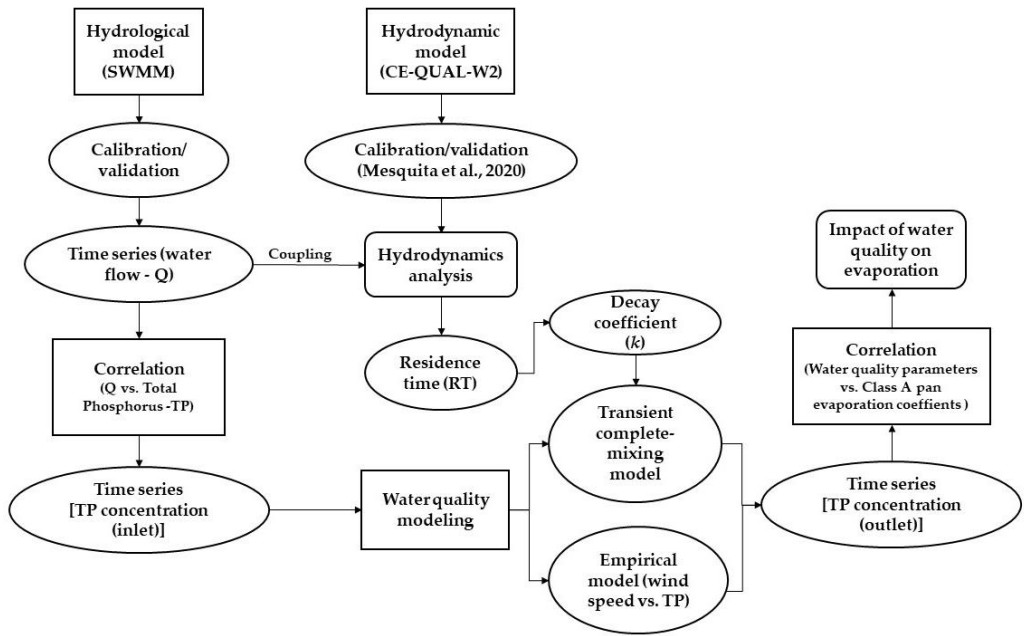

**Figure 2.** Flowchart of the study framework [10].

### 2.3. Hydrological and Hydraulic Modeling

For the hydrological and hydraulic modeling of the Santo Anastácio Lake basin, the computer model Storm Water Management Model (SWMM) was used, a software widely used for urban watersheds developed by the Environmental Protection Agency (EPA) of the United States of America (USA). SWMM is a one-dimensional hydrodynamic rainfall-runoff transformation model that uses the dynamic wave equations (Saint-Venant equations) and their simplifications (kinematic wave) for hydraulic simulation. The SWMM represents the hydrological processes over the basin (precipitation, infiltration, and runoff, for example), as well as simulates the hydraulic conditions of the flow in the conduits, such as the wave propagation and the effects of a backwater [27]. SWMM was selected for this work, as it is free and widely applied in several studies reported in the literature (e.g., [2,6,11,12]).

The computational base used in SWMM was initially conceived through geoprocessing in QGIS software version 3.22.6. The extraction of contour lines and the layout of the watershed were carried out using the image file (raster), called digital terrain model (DTM), with a precision of five meters, provided by the City of Fortaleza. With the contour lines, the point of discharge of the basin was initially defined, and the geoprocessing tools were applied to define the hydrographic basin of interest. In addition, the watercourses were vectorized and compared to the rainwater drainage network of the City of Fortaleza (galleries and channels), with a view to making the vectorized watercourses compatible

with the rainwater drainage network currently implemented. These procedures were carried out for a simplified representation and simulation of the waters of the drainage system due to the lack of consistent data on the microdrainage network (such as the height of the manholes, diameter of the galleries by sections, and others). In addition, the generated network helps in the process of optimizing the macro-scale analysis of drainage systems in large cities, using the tools of geographic information systems in view of the lack of data [2].

Then, the watershed was defined in the SWMM, matching the main watercourse (channel) and the main inflows to the east (galleries). For this purpose, the sub-compartmentalization of the Santo Anastácio Lake basin was performed in order to better represent the areas of influence of the basin related to the respective urban drainage galleries. Therefore, the SWMM modeling domain for the basin was divided into six sub-basins; eight junctions, to which the representative flow from the galleries flows; eight conduits that make up the tributary channel to the Santos Anastácio lake; a rain gauge (located on UFC's Pici campus); a storage reservoir representing the Parangaba lake; and, finally, an outlet, tributary to Santo Anastácio Lake. For the inclusion of the storage reservoir (Parangaba lake) in the SWMM, a bathymetric map was used [32]. From this, the elevation-area-volume curve was generated, and its parameters were inserted in the model, specifying the minimum level of the Parangaba lake in relation to the maximum depth (3 m).

The input parameters required by SWMM, among others, are the following for the basins: percentage of waterproofing (% imperv); Manning coefficient for waterproofing and permeabilization (N-imperv and N-perv, respectively); storage on impermeable and permeable surfaces (D-store-imperv and D-store-perv, respectively); and the soil infiltration method, of which there are four models available: Green Ampt, Modified Green Ampt, Horton and Curve Number (Curve Number—CN of the Soil Conservation Service—SCS, United States) [27]. In addition to these parameters, there is also the area of the basin and the average slope, these variables being defined in the process of delimiting the basins and in the adjustments made in the SWMM itself.

It is noteworthy that the infiltration model selected for this paper was the CN-SCS. The CN values vary on a scale from 0 to 100 (tabled values), defined according to the conditions of use and occupation of the basin, with 100 being the highest possible surface runoff. The value of 80 was set in SWMM. It was taken into account that all the flow would be totally directed to the conduits. Sub-basin slopes were adjusted by 2%; those of the conduits, according to the elevation of each node, compatible with the terrain elevations, both defined through the aforementioned computational base (DTM). The percentage of waterproofing of the basin was adjusted to 56%, in addition to the parameters of the Manning numbers for the surfaces of the basins (N-perv-%, N-imperv-%), which were 0.1 and 0.01, respectively, considering the current conditions of the same, observed through the high definition images provided by the PMF; Google Earth images; on-site visits; and in the calibration process. The roughness of the channel, defined by the Manning number (n), was 0.015, the suggested value for concrete [27], the channel covering material.

The dry weather contributions to SWMM were obtained through data from the flow measurement campaigns along the channel carried out in October 2013, as mentioned. In addition, these flows were compared to those measured during the other dry months. In this way, the measured flows were distributed proportionally in each junction located in the SWMM. Thus, the sum resulted in the total drained at the outlet during the dry period, guaranteeing the representativeness of the model.

Rainfall data from hydrological stations were used as input data for the hydrological model to transform rainfall into flow in SWMM. Prior to the SWMM calibration process, a sensitivity analysis of the previously mentioned hydrological parameters of the basin was performed manually. Subsequently, the model parameters were manually adjusted, and simulations were conducted on the same days corresponding to the measured flow data. The calibration and validation processes were performed by comparing the flow data measured at the inlet of Santo Anastácio Lake and that modeled by SWMM. The statisti-

cal analyzes conducted were the Nash–Sutcliffe efficiency coefficient (NSE), coefficient of determination ($R^2$), and mean deviation (MD) in relation to the perfect fit of the model. The data used for calibration were the flows measured in 2013 and 2014 and, for validation, in 2018 and 2019. It is important to note that the calibration consisted of manually adjusting the model parameters, as already mentioned. The simulated flows obtained from the hydrograph were compared to the flows measured at the corresponding time. The time interval defined for obtaining the hydrographs in this process was 15 minutes. The hydraulic model selected for the simulations in this work was the dynamic wave model, the most complete one in SWMM.

### 2.4. Hydrodynamic Modeling of the Lake

CE-QUAL-W2 was the model selected for hydrodynamic modeling in Santo Anastácio Lake. This model was calibrated and validated in a previous study [10]. It is a two-dimensional (2D) hydrodynamic model that considers the longitudinal and vertical variations, despite the lateral variations. Models with the aforementioned characteristics are ideal for application in water bodies with great length in relation to width [28], such as Lake Santo Anastácio, being widely applied in the study of lakes and reservoirs (e.g., [15,16,33,34]). A two-dimensional model was chosen in this work, as it is possible to analyze the vertical and longitudinal profile of the lake in terms of hydrodynamic characteristics.

The CE-QUAL-W2 model requires bathymetric information, inlet and outlet flows, and meteorological and water quality data [28]. The lake under study has an approximate length of 900 m and an average width of 185 m, as already mentioned. Therefore, it was discretized, according to the bathymetry variation, in 32 longitudinal segments, with 29 m, and in vertical layers with a distance of 0.2 m per layer.

Subsequently, the measured data required by the model were inserted, such as: meteorological [air temperature (°C), dew point temperature (°C), speed (m·s$^{-1}$) and wind direction (degrees) and cloudiness (scale from 0 to 10)], obtained through the meteorological station; inflow and outflow temperature (°C), measured in the field; inlet flow (m$^3$·s$^{-1}$), modeled by SWMM; and outflow from the lake (m$^3$·s$^{-1}$), obtained through a linear correlation between the data measured in 2013 (inlet and outlet of the lake) and extrapolated with the flow data simulated by SWMM for all the years analyzed. In addition, geographic coordinates, bottom elevation, initial water temperature (28 °C), and sediment temperature (29 °C) were adjusted. Regarding the hydraulic parameters, viscosity, and wind roughness, the standard values of the CE-QUAL-W2 version 3.7 model were used [28]. Friction was calculated according to the Chézy equation.

CE-QUAL-W2 simulates hydrodynamics and heat transport, starting by calculating the water surface elevation (free surface equation). Then solve the equations of conservation of mass and momentum (hydrodynamics). Subsequently, it calculates the heat exchanges (energy balance equation). Finally, it simulates the heat transport equation (2D transient regime), from which it is also possible to calculate the specific mass of water (equation of state) [28].

The calibration and validation of the parameters of the CE-QUAL-W2 model were carried out by statistical analysis, comparing the calculated and modeled evaporation data through the mean deviation and coefficient of determination ($R^2$). It is noteworthy that the evaporation calculated with the 2018 data was used to validate the model. After the calibration and validation of the CE-QUAL-W2 model, it was possible to perform hydrodynamic and evaporation simulations in the lake. Water temperature measurements were also presented to validate the vertical temperature profiles of the CE-QUAL-W2 simulations, with a time step of six hours [10].

### 2.5. Integrated Basin and Lake Analysis

The influence of the hydrological characteristics of the Santo Anastácio Lake watershed and its impact on hydrodynamics, water quality, and, hence, on evaporation, was analyzed. To this end, three different years of a time series with 20 years of data (2000 to 2019) were

selected by calculating the deviation from the average: with rainfall above the historical average (2019), below the average (2013), and a year close to the average (2018), considered typical. Therefore, simulations were performed in SWMM for 2013, 2018, and 2019.

Thus, the computational base of SWMM and CE-QUAL-W2 was designed for 2013, 2018, and 2019 and coupled. The coupling process consisted of simulating rainfall during those years (2013, 2018, and 2019) in order to reproduce the hydrological conditions of the basin in the SWMM. Subsequently, time series of flows were generated for the corresponding periods, which were entered as input data in CE-QUAL-W2. The time interval defined for the simulation in SWMM, in this case, was hourly during each corresponding year. In this way, it was possible to analyze the hydrodynamic processes in Santo Anastácio Lake, considering the different hydrological responses arising from its contribution basin.

The hydrodynamic patterns for the years considered dry, typical, and rainy (2013, 2018, and 2019, respectively) were analyzed through the following variables: thermal stratification, horizontal velocity, and hydraulic residence time (RT). Different depths were verified, such as at the water surface, at 2.8 m and 5 m, and compared to the complete mixing regime. From this, time series of the variables under study were generated to analyze the influence of the hydrological regime on the hydrodynamic patterns of shallow reservoirs. Furthermore, with the values of the hydraulic residence time for the mentioned years, the decay coefficient of phosphorus in the lake, $k$ (year$^{-1}$), was also evaluated. To this end, the model shown in Equation (1) [24] was applied, which was validated by several authors [4]. The $k$-values obtained were also used in the complete-mix transient water quality model [35], later applied for several studies [36,37], as described in Section 2.6.

$$k = \frac{4}{\sqrt{\text{RT}}} \tag{1}$$

*2.6. Water Quality Modeling and Evaporation*

Time series of TP concentration of Santo Anastácio Lake were generated for 2013, 2018, and 2019. Initially, the measured inflows were correlated with the TP concentration in 2013, 2014, 2018, and 2019. The coefficient of determination was evaluated as a basis for choosing the function with the best fit to the data. Through the adjusted function, the flows generated in SWMM for the years of interest were used. In this way, the TP concentrations of water inflow to Santo Anastácio Lake were obtained.

Subsequently, two water quality models were applied and compared to obtain the concentration at the outlet of the lake: the transient model of complete mixing [24,35], later validated [36], Equations (2) and (3); and the empirical model that relates average wind speed, $v_m$ (m·s$^{-1}$) and TP (mg·L$^{-1}$) [10], as shown in Equation (4).

$$\text{TP}(t) = \text{TP}_o e^{(-\omega t)} + \frac{w}{V\omega}\left(1 - e^{(-\omega t)}\right) \tag{2}$$

$$\omega = \frac{Q_s}{V} + k \tag{3}$$

$$\text{TP} = 0.8435 v_m + 1.0735 \tag{4}$$

where: TP($t$) = total phosphorus at time $t$ (mg·L$^{-1}$); TP$_o$ = total phosphorus at time $t -$ 1 (mg·L$^{-1}$); $t$ = time step (year); $w$ = phosphorus load at the entrance to the water body (g·year$^{-1}$); $V$ = volume (m$^3$); $Q_o$ = outlet flow (m$^3$·year$^{-1}$); and $k$ = phosphorus decay coefficient (year$^{-1}$). It is noteworthy that the $k$-values applied in Equations (2) and (3) were the same as those previously calculated [24] (Equation (1)). The volumes ($V$) were obtained through the water balance performed by the CE-QUAL-W2 model. The outflow ($Q_o$) was obtained through the correlation between the inflow and outflow mentioned above using the outflows generated by the SWMM model as an input variable to obtain the time series of outflows.

Finally, the time series of modeled TP concentration was correlated with the monthly Class A pan coefficients obtained [11]. This evaluation was intended to expand the data

series in order to better verify the impact of water quality on evaporation rates. Thus, the rainy and dry periods were segregated, and the influence of climatic seasonality on water quality and, consequently, on evaporation rates was verified. Additionally, a correlation was made between the measured data of BOD with the Class A pan evaporation coefficients for 2013, the year with data availability. This correlation was carried out in order to evidence the impact of water quality on evaporation rates using another water quality parameter.

### 2.7. Scenarios

Scenarios were analyzed considering the null flow to Santo Anastácio Lake during the dry season. It was assumed that the contribution of dry weather comes completely from sanitary sewage. Therefore, if there were interventions in the basin, such as the implementation of adequate infrastructure for the adequate collection, transport, and disposal of sewage, the flow would be null or minimum during the dry season. It is important to note that this analysis was carried out with a view to identifying the possibilities of direct interventions in Santo Anastácio Lake for its environmental recovery, as well as to minimize possible structural risks to the dam since it is an artificial lake.

Thus, the average water level reduction in the lake was simulated, considering the evaporation losses equivalent to those modeled by CE-QUAL-W2, measured by the Class A pan and by the Piché evaporimeter. Thus, it was possible to obtain the minimum water level in the lake, in which case direct interventions for environmental recovery could be carried out.

## 3. Results

### 3.1. Hydrological Model and Analysis of Hydrological Variability

Figure 3 presents the computational base in SWMM for the Santo Anastácio Lake basin in Fortaleza, Ceará, Brazil.

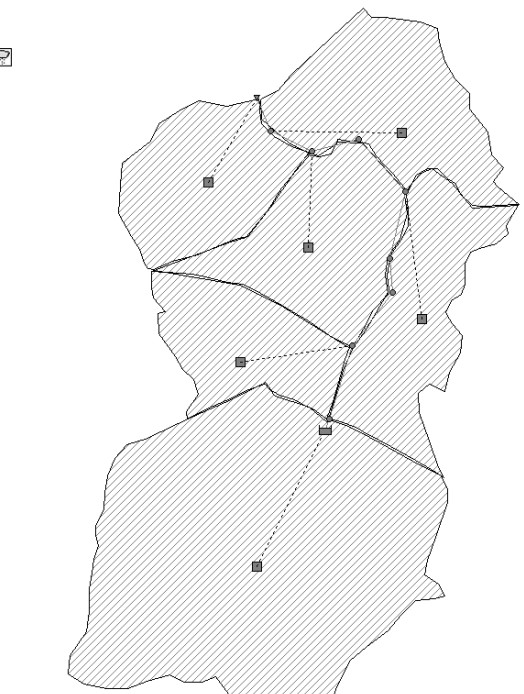

**Figure 3.** Computational base in the SWMM of the basin of the Santo Anastácio Lake inlet section in Fortaleza, Ceará, Brazil.

Figure 4a presents the result of the calibration using the flow data measured and modeled by SWMM in 2013 and 2014. Through the calibration of the model parameters, an NSE coefficient of 0.86 was obtained, an $R^2$ coefficient of 0.96, and a mean deviation

of 29%. Figure 4b presents the result of the SWMM validation process, with data from 2018 and 2019, for the same statistical variables, resulting in an NSE coefficient of 0.73, an $R^2$ coefficient of 0.94, and an average deviation of 38%. Previous surveys performed with SWMM obtained similar NSE values: 0.616–0.899, in Orissa, India [11]; 0.71–0.75, in Lille, France [12]; 0.76–0.78, in Helsinki, Finland [38]; and 0.93, in Fortaleza, Brazil [2]. Regarding the mean deviation, the values obtained here were within the range reported in the literature of 7–54% [2].

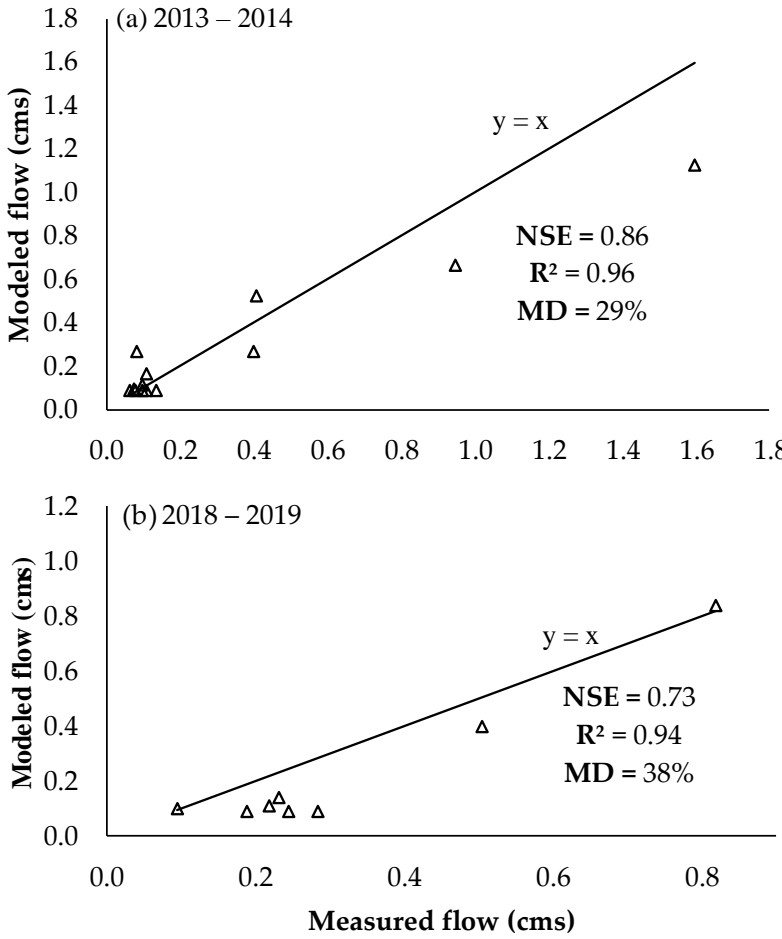

**Figure 4.** Calibration with flow data from (**a**) 2013 and 2014 and (**b**) validation with flow data from 2018 and 2019 of the SWMM model in the Santo Anastácio Lake basin.

Therefore, the present work presents values within the ranges reported by the literature, attesting to the representativeness of the rainfall-runoff model used for the lake basin under study. It is worth mentioning that the measured flow values obtained are subject to the variability of the hourly water consumption and, consequently, to the release of sanitary sewage into the drainage network since there are several unauthorized sewage connections in the drainage network inflow to the Santo Anastácio Lake [3,5], as mentioned. In addition, there are numerous uncertainties inherent to hydrological studies related to the variability of flows, notably in urban basins [2]. This fact may contribute to the increase in the deviation between measured and modeled data.

In the evaluation of the time series of flows generated by the SWMM, it is verified, as expected, that the total inflows increase according to the total annual precipitation, being the lowest in 2013, followed by 2018, with the highest values in 2019, following the pattern of the region [5]. Furthermore, higher flows into the lake are observed predominantly in the first half of the year, resulting from temporally concentrated rainfall, especially from January to May. In the second half of the year, the channel inflows are mostly derived from

the contribution of the release of sanitary effluent to the galleries and channel, with average values of approximately 120 L·s$^{-1}$ [3,4].

These patterns are also consistent with those reported in measurements performed in the same study area in 2014 [39] and 2018 [5].

### 3.2. Inflow and Outflow

The correlation between the flow data measured at the inlet and outlet of Santo Anastácio Lake in 2013, which was used to generate the time series of outflows in Santo Anastácio Lake for all the years analyzed in this work (2013, 2018, and 2019), resulted in an $R^2$ of 0.99. Therefore, the representative correlation of the patterns between the inflows and outflows to the lake was considered. This fact demonstrates the effectiveness of using the time series generated, from the correlation, as input data for the CE-QUAL-W2 model. It is noteworthy that the average difference between the inflow and outflow of Santo Anastácio Lake is approximately 8%, corroborating a previous study in the same lake [5], which is attributed to evaporation losses.

### 3.3. Hydrodynamic Modeling

3.3.1. Impact of Hydrological Variability on Hydrodynamics

Figure 5 presents a synthesis of the coupling between SWMM and CE-QUAL-W2, illustrating (a) the annual series of precipitation and flow generated in the SWMM and (b) the time series of temperature according to depth, in Santo Anastácio Lake, resulting from hydrodynamic modeling using the CE-QUAL-W2 model, in 2018, as a demonstration of the general pattern.

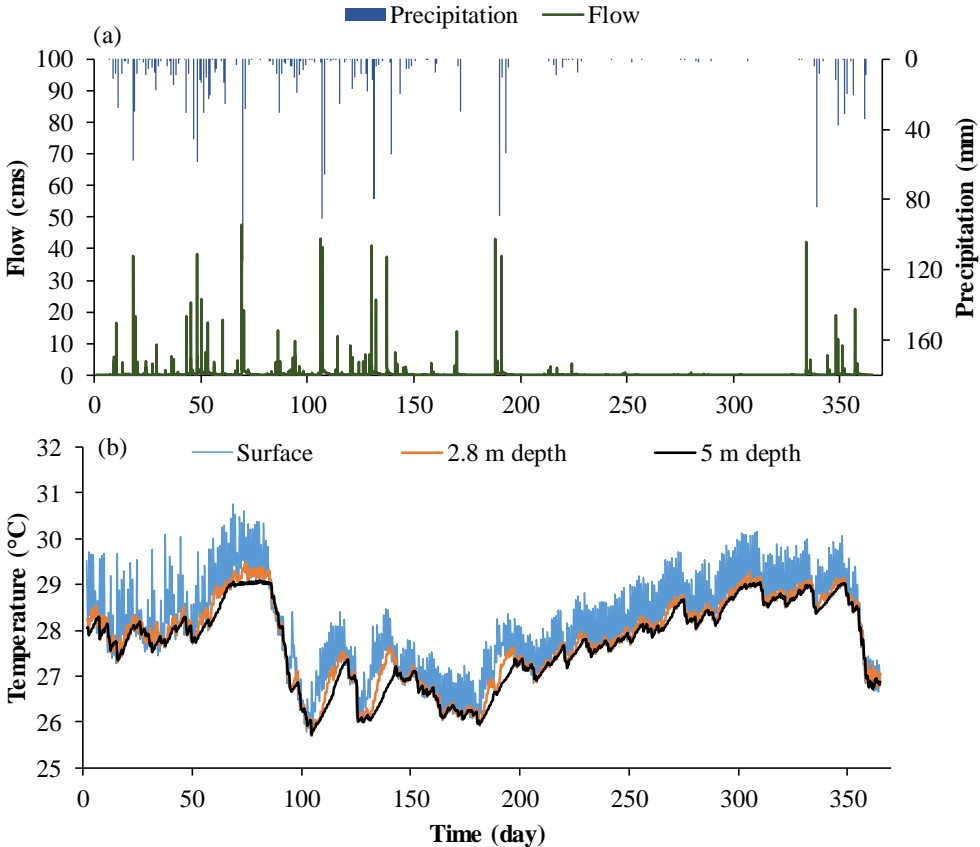

**Figure 5.** Synthesis of the coupling between the SWMM model and the CE-QUAL-W2 through the (**a**) annual series of precipitation and flow generated in the SWMM and the (**b**) temperature distribution in the Santo Anastácio Lake water column at different depths (surface, 2.8 m and 5 m) in the year 2018, typical.

It can be seen that the daily variation in temperature between the surface and the deepest layer (5 m) tends to maintain the general pattern practically throughout the year, with daily thermal stratifications of up to 2 °C. As expected, the temperature decreases with increasing depth (2.8 m and 5 m), and the influence of seasonality is evident. The water temperature in the lake tends to be higher in the rainy season (average of 28.2 °C), decreasing in July, in view of the lower air temperatures characteristic of this period; and gradually increases in the dry season due to the higher air temperatures observed in the region [10]. However, lake temperatures in the dry season (average of 27.7 °C) are still slightly lower than those in the rainy season for the year 2013 (dry year), denoting the predominance of the influence of the inlet flow on the lake temperature pattern. In the typical year (2018), the water temperature in the dry season is higher (28.16 °C) than in the rainy season (27.57 °C), following the trend of air temperature. In 2019 (rainy year), there are no significant differences between water temperatures in the dry (28.51 °C) and the rainy season (28.48 °C).

It is known that the wind speed is higher in the dry season [10], which possibly contributes to the reduction in temperature due to the mixing process of the water column [20]. In this sense, the temperature can affect the general circulation pattern of the lake [14] and, consequently, the other physical, chemical, and biological properties of the environment under study [16,22], despite the shallowness of the lake. Additionally, it may point to the need to understand the thermal distribution of the water column, notably in deeper reservoirs, such as those found in the semiarid region of Northeast Brazil [19]. Reservoirs subject to long periods of drought, with lower depths, tend to have lower thermal stratifications and, consequently, may present a polymythic pattern. That is, with more frequent mixing events of the water mass [10,19,20]. Water temperature is also subject, in the long term, to the effects of climate change [21,22], which can severely impact the biotic environment [23].

Furthermore, the inflow into the lake affects the horizontal water velocity, as expected, averaging across the reservoir between 0.002 and 0.001 m·s$^{-1}$ during the rainy and dry seasons, respectively. However, no significant differences were identified in the interannual mean horizontal velocity.

### 3.3.2. Hydraulic Residence Time

The average residence time is 34.86, 34.13, and 34.10 days for 2013, 2018, and 2019, respectively. It can be seen that the average residence time tends to decrease slightly with the increase in annual precipitation but remains practically constant inter-annually. In addition, it is observed that the average residence time in 2013, for example, varies between 16.67 and 36.67 days, an average of 26.77 days, and 14.68 and 57.74 days, an average of 39.97 days, in the first and second semesters of the analyzed years, respectively. These values are within the ranges reported in the literature [15]. Thus, the influence of seasonality, notably precipitation, on the lake circulation pattern is evident, as expected. Furthermore, there is a small difference between the RT of the deeper layers of the lake (2.8 and 5 m) compared to the RT of the surface (average of 3%). When comparing all layers (surface, 2.8 and 5 m) to the RT of the complete mixing regime, a difference of 1% and 2%, respectively, is observed. Therefore, the predominant factor for the variation of RT values of shallow lakes is the variability of inflows [24], probably in view of the small depth. On the other hand, in deeper reservoirs, the inflow can also be a determinant for the values of RT. The higher the inlet flow, the lower the RT tends to be [16,24]. A previous study [16] showed low RT values in a deep reservoir, in Pará, Brazil, possibly due to large inflows. Thus, the hydrological variability is crucial for the general pattern of water circulation and, therefore, for the reduction of the hydraulic residence time, as was also observed in Santo Anastácio Lake.

Regarding the phosphorus decay coefficient (*k*), there is an average value of 14, 15, and 18 year$^{-1}$ for 2013, 2018, and 2019, respectively. The values obtained corroborate the order of magnitude of 20 year$^{-1}$ calculated for the Santo Anastácio Lake [4]. It can also be observed that the rainier the year, the greater the variability of the coefficient *k* in the first

semester of the year, although still relatively small. Thus, the seasonal coefficients obtained were 15.01, 19.87, and 26.88 year$^{-1}$ in the rainy season and 12.8, 11.76, and 11.90 year$^{-1}$, in the dry period, for 2013, 2018, and 2019, respectively.

Thus, the seasonal *k* coefficients are presented in an increasing way, notably during the rainy season, with little variability in the dry period of the three years. This slight variability in the values of the *k* coefficients can be explained due to their dependence on the hydraulic residence time and, therefore, on the inlet flow [4,16,24].

### 3.4. Impact of Inlet Flow on Water Quality

Figure 6 shows the correlation between the inflow to Santo Anastácio Lake and the phosphorus concentration at the inlet. It is verified that the fitted function was a power law, with an R$^2$ of 0.70; therefore, statistically significant. It is noteworthy that this correlation was used, as mentioned, to generate the time series of concentration of phosphorus inflowing to Lake Santo Anastácio in 2013, 2018, and 2019. It is observed that the higher the flow, the lower tends to be the phosphorus concentration; however, tending to stabilize at approximately 0.8 mg/L. This fact could be attributed to the process of dilution of the pollutants carried, mainly phosphorus, the greater the surface runoff generated in the rainiest months. However, in the dry season, with lower flows, high concentrations are maintained due to contributions from sanitary sewage and/or solid waste discharged into the influent channel [4]. It is important to mention that the concentrations of total phosphorus in raw sewage are normally in the range between 5 and 25 mg·L$^{-1}$ [40].

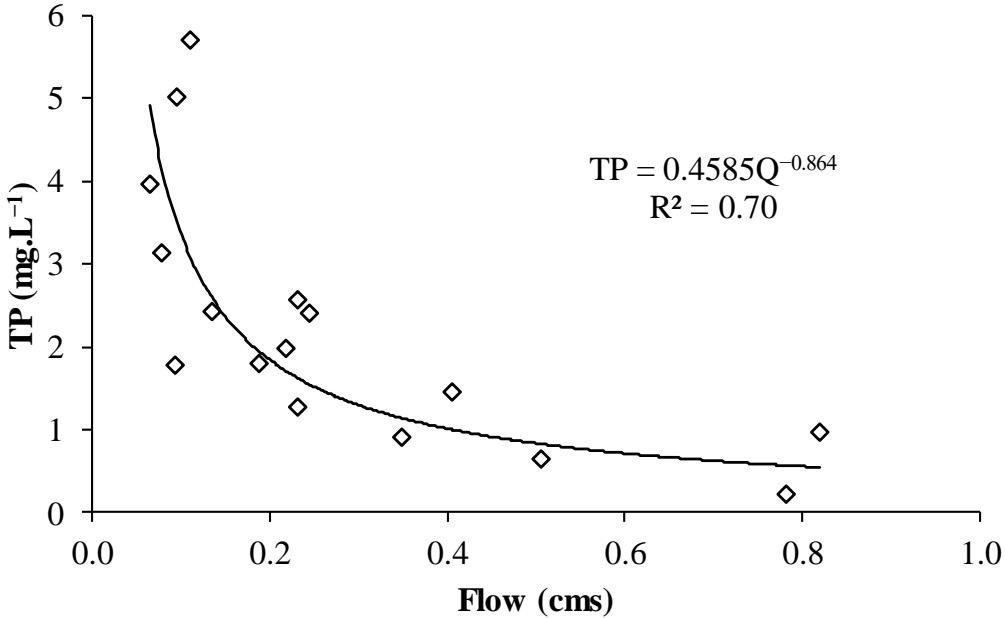

**Figure 6.** Correlation between flow (Q) and total phosphorus concentration (TP) in tributaries to Santo Anastácio Lake.

### 3.5. Impact of Water Quality on Evaporation Rates

Figure 7a,b shows the phosphorus concentrations calculated at the Santo Anastácio Lake outlet, using the complete-mix transient water quality model [24,35] and the empirical model [10], based on wind speed, in (a) 2013, (b) 2018, and (c) 2019. It is verified that the TP concentration values generated with the complete-mix model tend to be higher than those of the empirical model in all the years analyzed. On the other hand, a similar trend can be observed between the models, despite the differences between the physical principles of the models. Therefore, both models point to the representativeness of lake water quality patterns.

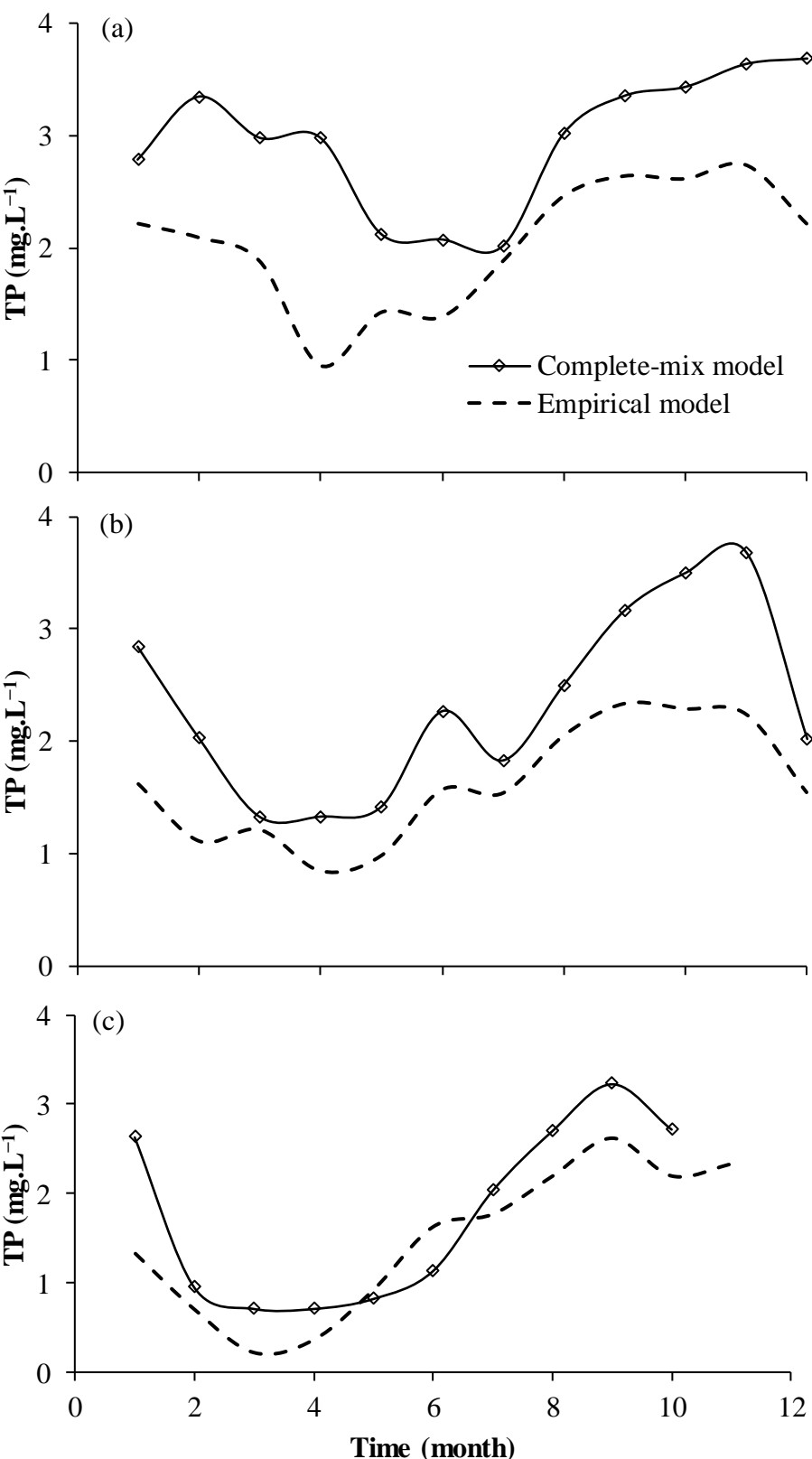

**Figure 7.** Time series of total phosphorus concentration (TP) (mg·L$^{-1}$), generated from the complete-mix model and the proposed empirical model (TP $= 0.8435v + 1.0735$, based on wind speed, in (**a**) 2013, (**b**) 2018, and (**c**) 2019, at Santo Anastácio Lake in Fortaleza, Ceará, Brazil.

The time series of TP concentration shown in Figure 7a–c were correlated to the Class A pan coefficient, as shown in Figures 8 and 9a,b. The results indicate similar patterns as those previously observed [10].

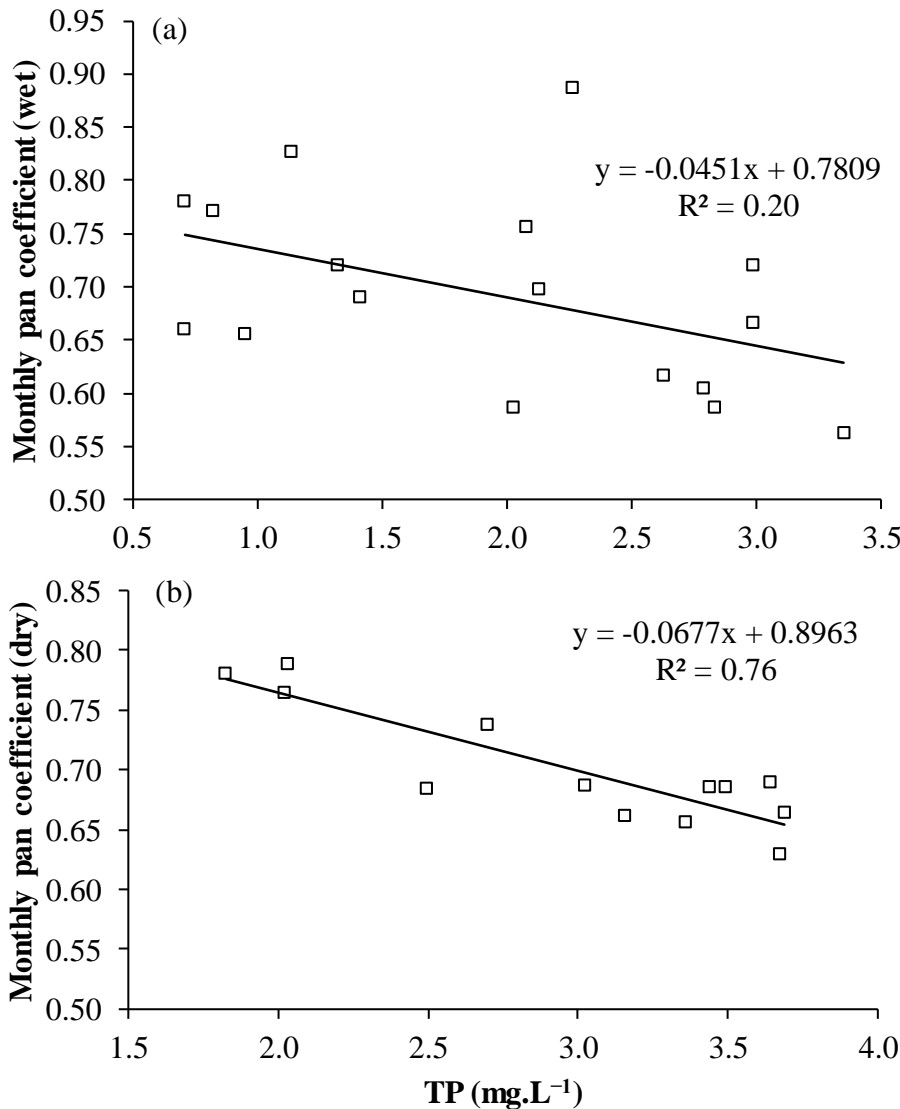

**Figure 8.** Linear regression between the monthly average of total phosphorus concentration (TP) (mg·L$^{-1}$), generated by the complete-mix model, and the monthly Class A pan coefficient for (**a**) rainy and (**b**) dry seasons from 2013, 2018, and 2019.

The linear correlation in Figure 8a,b, which shows the TP concentrations obtained through the complete-mix model, showed an $R^2$ of 0.20 and 0.76 for the rainy and dry seasons, respectively. Figure 9a,b, which contains the TP concentrations obtained by the empirical model based on wind speed, showed an $R^2$ of 0.048 and 0.52 for the rainy and dry seasons, respectively. It is clear that the dry period presented statistically significant values using both models, demonstrated through the high values of $R^2$ ($R^2 > 0.5$). On the other hand, in the rainy season, there was no strong correlation between TP concentrations and the Class A pan coefficients. This can be explained by the lower TP concentrations observed in this period [4]. In addition, it is also inferred that it is due to the greater variability of temporal conditions. This could cause changes in the measurements obtained in the Class A pan and, combined with variations in water quality due to precipitation, would result in patterns that are more difficult to be explained.

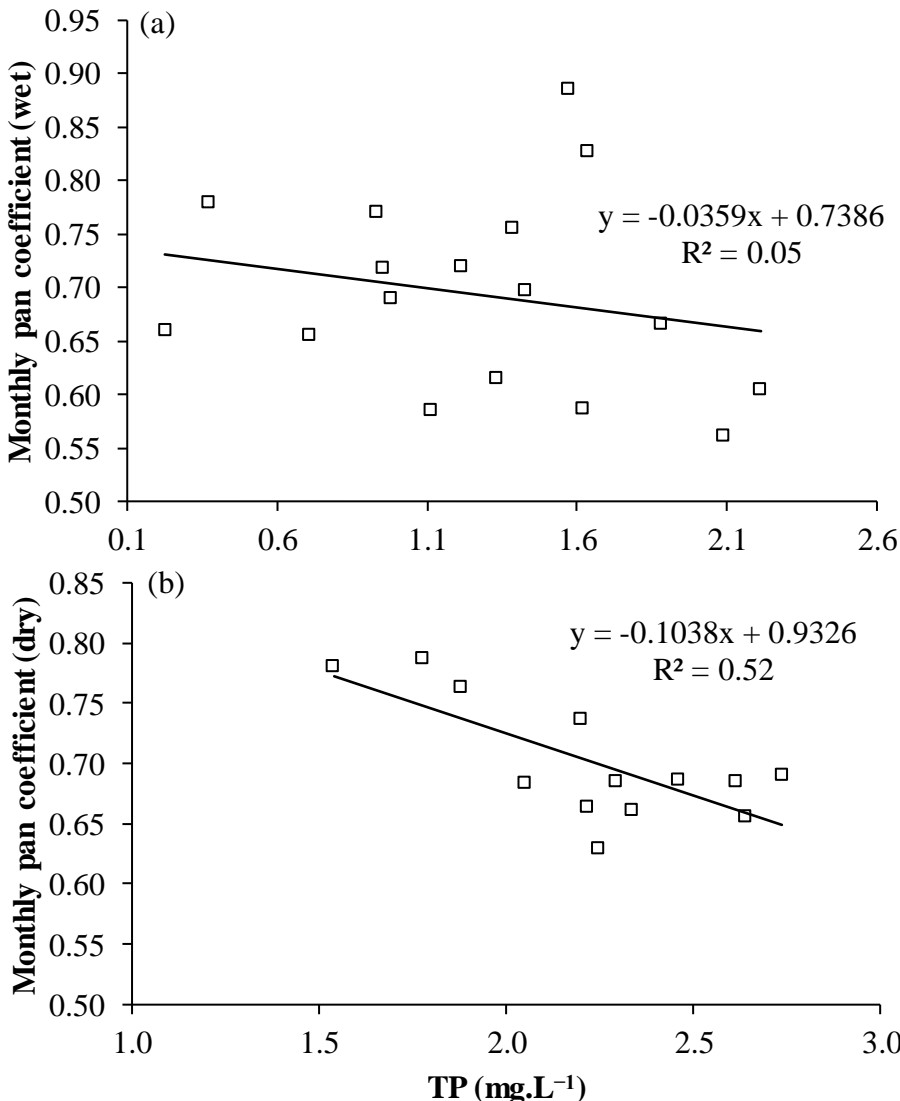

**Figure 9.** Linear regression between the monthly mean concentration of total phosphorus (TP) (mg·L$^{-1}$), generated by the proposed empirical model, and the monthly Class A pan coefficient for the (**a**) rainy season and (**b**) the dry season of 2013, 2018, and 2019.

Figure 10 shows the correlation between the BOD and the monthly Class A pan coefficients. As shown, the linear regression showed a negative trend, with an $R^2$ of 0.85, therefore, statistically significant. Thus, the trends observed using TP are confirmed. It is known that BOD is an indicator of the presence of organic matter in water [40]. In this case, high BOD values can also come from organic loads from sanitary sewage and/or solid waste [4].

It is important to mention that a quantified portion of BOD can also be present in water in the form of dissolved organic carbon (DOC) [40]. In this sense, it corroborates with other studies [8,9] that observed a relationship between the high concentrations of DOC and the reduction of the overall heat content of the studied lakes. This could impact the reduction of evaporation rates [10]. On the other hand, another study [41] pointed to air pollution as a possible effect of reducing evaporation in reservoirs. Thus, there are several anthropic forcings that possibly affect hydrological variables, notably evaporation. It is important to highlight that all these topics need deeper studies to understand their relationship with evaporation in natural environments.

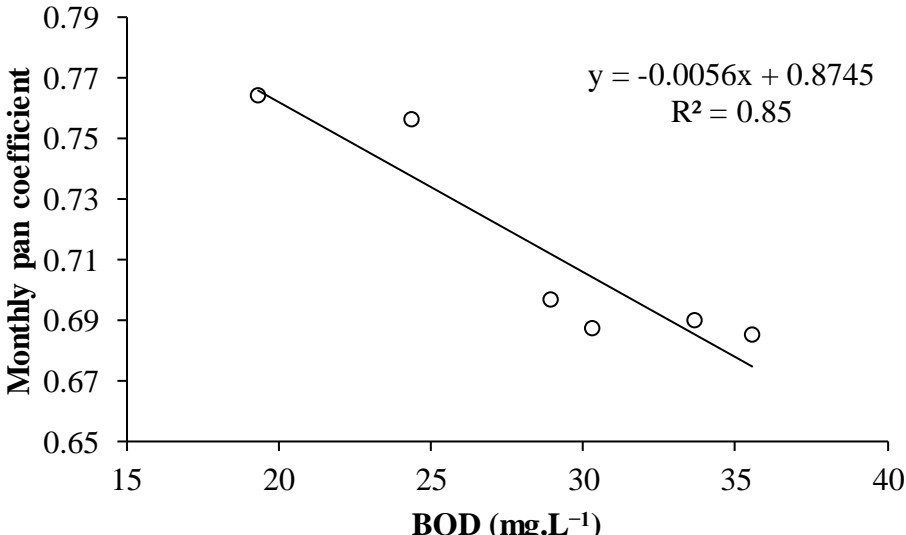

**Figure 10.** Linear regression between biochemical oxygen demand (BOD) and monthly Class A pan coefficient for the year 2013.

*3.6. Scenario Analysis*

Figure 11a,b presents the results of the simulations of the reduction of the average level of Santo Anastácio Lake in the dry period, considering the zero inflow and different values of evaporation for 2013 and 2018. Whether the lowest water level, in this case, null, would be obtained by simulating the water balance considering the evaporation values of the Class A pan; followed by evaporation modeled by CE-QUAL-W2 model (approximately 0.4 m); and, finally, with the evaporation of the Piché evaporimeter (0.6 m, approximately). The analysis of these water balance scenarios, as mentioned, is necessary to propose possible alternatives for direct interventions in the Santo Anastácio Lake hydraulic basin, aiming for its environmental recovery. The possibility of a minimal or zero inflow to the lake could be considered in the case of the implementation of an adequate sanitary sewage system throughout the basin in order to avoid the direct release of sanitary sewage into the drainage network.

It is known that Santo Anastácio Lake has reduced its storage capacity over the years due to silting, which is normally more intense in urban basins [42]. Additionally, the deterioration of water quality, with the accumulation of macrophytes, can increase the structural risks of dam failure. These factors increase the need for direct interventions with a medium to long-term outcome horizon.

There are some traditional alternatives for restoring the reservoir and environmental capacity of lakes, such as dredging the bottom sediments [43]. However, this alternative, in addition to being costly, can induce the resuspension of materials, sometimes toxic, deposited on the bottom [43,44]. In this sense, this practice may have the effect of reducing water quality, solving the problem only partially [43]. Therefore, the analyzed scenarios of water level reduction, considering a medium to the long-term perspective of interventions in the basin, aim to subsidize the proposition of solutions for environmental recovery [45,46].

One of the alternatives could be the removal of the bottom sediment during the dry season, similar to what has been proposed in semiarid reservoirs [47]. Although the removal of bottom sediments by excavation is also an expensive technique, this practice could improve water quality, as it would reduce the load of nutrients, notably phosphorus, which can return to the water column due to resuspension induced by the wind and/or through the resolubilization process under conditions of low dissolved oxygen concentration [48]. However, it is important to emphasize that, in the study reported in the literature [47], the objective is to use the sediment for agricultural purposes. On the other hand, in Santo

Anastácio Lake, it is inferred that the short-term perspective is just removal for proper disposal. In this case, because it is an urban source, possibly more subject to potentially toxic contaminants [44,49]. In addition, to recover the water quality in Lake Santo Anastácio, it is recommended that sanitation conditions be improved in its basin, avoiding, for example, the release of sewage into the drainage network.

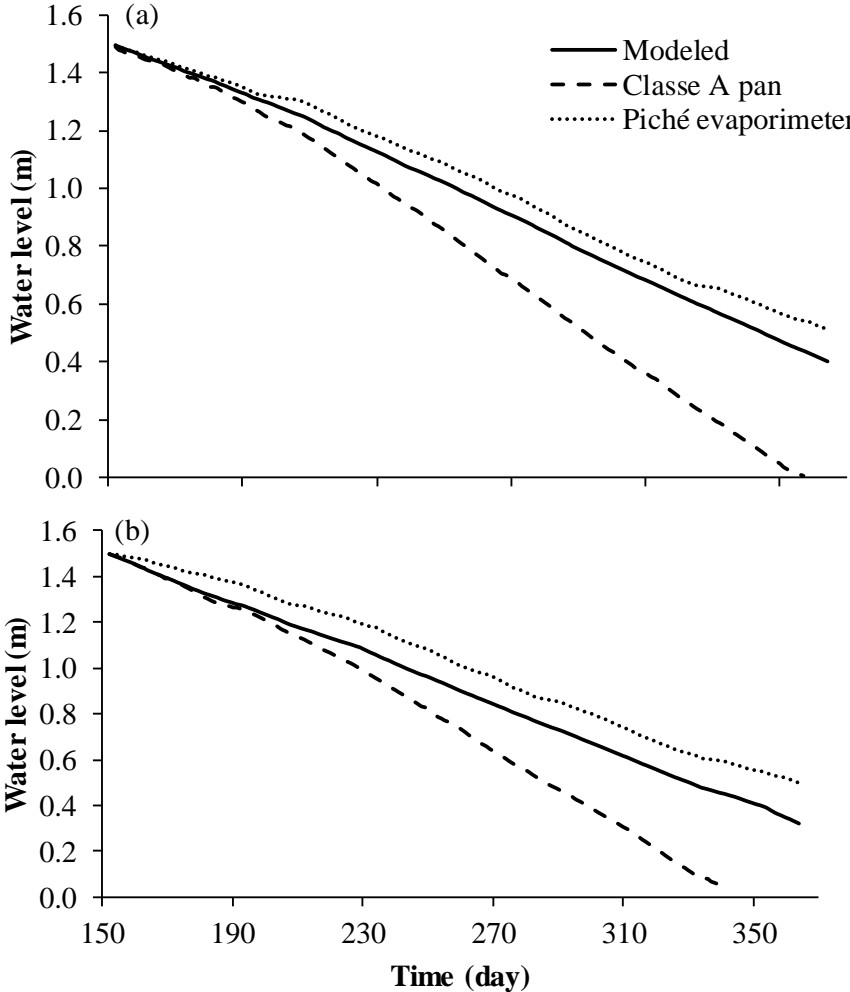

**Figure 11.** Simulation of the reduction in the average level of Santo Anastácio Lake, in the dry season, considering the zero inflow and different evaporation values, for the years (**a**) 2013 and (**b**) 2018.

It is also noteworthy that this scenario analysis is only the starting point for environmental recovery proposals that must be evaluated in a specific way.

## 4. Conclusions

In the present study, integrated basin-lake modeling was carried out to evaluate the impact of hydrology on the hydrodynamics of a shallow lake, considering water pollution and its impact on evaporation rates. By coupling SWMM to CE-QUAL-W2, a slight variation of the hydrodynamic characteristics was observed with the increase in annual precipitation, such as an increase in the average water temperature, a decrease in hydraulic residence time, and an increase in the phosphorus decay rates. Seasonal variations were also verified from the rainy to the dry periods.

Regarding the analysis of water pollution, statistically significant correlations were observed between the inflow and the concentration of total phosphorus. This fact indicates the influence of the input loads from the basin, with the effect of phosphorus dilution prevailing in the rainy season.

The analysis of the impact of water pollution on evaporation rates, applying a complete-mix water quality model and an empirical one based on wind speed, confirmed the negative and statistically significant linear regression trends between the total phosphorus concentration and the monthly Class A pan coefficients, especially in the dry season. A similar trend was also observed when monthly Class A pan coefficients were correlated with BOD. Thus, the influence of pollution on the reduction of evaporation rates was evidenced.

Finally, a simple application of the results was performed to illustrate the impact of different evaporation rates on water level reduction, and potential environmental restoration measures were discussed. This study is important to improve the management of lakes and reservoirs by including the impact of pollution on the water balance.

**Author Contributions:** J.B.d.F.M.: Conceptualization, Funding acquisition, Data curation, Formal analysis, Methodology, Writing—Original draft preparation, and Software; I.E.L.N.: Conceptualization, Funding acquisition, Methodology, Supervision, and Writing—Reviewing and Editing. All authors have read and agreed to the published version of the manuscript.

**Funding:** The present study was supported through the Coordination for the Improvement of Higher Education Personnel—CAPES (research grant PROEX/2022).

**Institutional Review Board Statement:** Not applicable.

**Informed Consent Statement:** Not applicable.

**Data Availability Statement:** Available upon request.

**Conflicts of Interest:** The authors declare no conflict of interest.

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
