# Peer review of "Coupling Hydrological and Hydrodynamic Models for Assessing the Impact of Water Pollution on Lake Evaporation"

_sustainability, doi:10.3390/su142013465_

Round 1

Reviewer 1 Report

As per my view the study is robust and falling under the scope of the journal. The results from this study are important to assist in the operational management of lakes and reservoirs, including the impact of pollution on the water balance. Before accepting for publication manuscript need minor cum major revision as suggested: 

Line no 66, 68: authors used multiple references, not used in scientific aspect, kindly used the latest one or reference which is very important 

Figure 1: latitude and longitude need to be given in scientific formate 

Abbreviations once extended kindly use (e.g., BOD, DO, TP, Curve number)

Line no 533-538: text need to be supported with recent references (Arab J Geosci 9: 28." doi. org/10.1007/s1251 (2015): 7-015; Arab J Geosci 10, 134 (2017). https://doi.org/10.1007/s12517-017-2915-2)

Line no 540-544: need to be supported with reference (Water Resour. Manag. 6, 36 (2020). https://doi.org/10.1007/s40899-020-00396-6)

Conclusion need to be re-written in fruitful ways

Author Response

Dear Reviewer,

Reviewer 2 Report

It is an interesting paper that sets out to evaluate the impact of hydrological variability on hydrodynamics, considering water quality and its impact on evaporation rates in an urban tropical lake. The paper combines field information and desk information.

In general, the paper can be published with some minor revisions as below:

1- I suggest adding a sketch for the methodology to be more clear and more straightforward.

2- A sensitivity analysis is missing for the hydrological parameters. 

3- The conclusion is not well supported by the results.

Author Response

Dear Reviewer,

Reviewer 3 Report

Dear Authors,

Congratulations! I would like to thank the author team for their excellent effort on this fascinating research about analyzing the effects of hydrological variability on hydrodynamics while taking into account the effects of water quality on evaporation rates in an urban tropical lake. The lake basin was modeled using the hydrological-hydraulic Storm Water Management Model (SWMM), and the hydrodynamics and direct evaporation of the lake were modeled using the two-dimensional CE-QUAL-W2 model. In order to perform the integrated basin-lake modeling, the two models were connected.

A few things need to be clarified.

-        What is the author's proposed research hypothesis?

-        The SWMM and CE-QUAL-W2 models used in the study: were they created by the author or were they models that had been calibrated and validated? If so, kindly describe the calibration and validation (process and) findings of the model.

-        Do you need further information on choosing rainfall values for sub-catchments? What about infiltration and estimation methods?

-        Why is the use of a two-dimensional model required? Therefore, the author must be clear about the kind of data that is supplied into the two-dimensional model.

-        The locations of the hydrological stations were utilized to calibrate and validate the model?

-        The article's use of the SWMM model and the CE-QUAL-W2 model has to be properly explained, and the author needs to include a flowchart (input data, simulation, calibration, validation, and output of data).

-        Figuring 10: Did evaporation alone cause the lake's water level to fall? no inflow or outflow from the lake? What recommendations do the authors have for the sustainable management of water resources or water quality in the study area? These need to be made clear by the authors.

Some minor points need to be fixed:

- The abstract needs to be changed to be shorter.

- Figures 1 should also show the direction of the wind.

Thank you.

Author Response

Dear Reviewer,
